# Cost of Surgical Treatment of Obesity and Its Impact on Healthcare Expense–Nationwide Data from a Polish Registry

**DOI:** 10.3390/ijerph20021118

**Published:** 2023-01-08

**Authors:** Magdalena Osińska, Iwona Towpik, Yaroslav Sanchak, Edward Franek, Andrzej Śliwczyński, Magdalena Walicka

**Affiliations:** 1Department of Internal Diseases, Endocrinology and Diabetology, National Medical Institute of the Ministry of the Interior and Administration, 137 Wołoska Str., 02-507 Warsaw, Poland; 2Department of Internal Diseases, Diabetology, and Endocrinology, Collegium Medicum, University of Zielona Góra, 28 Zyty Str., 65-046 Zielona Góra, Poland; 3Department of Human Epigenetics, Mossakowski Medical Research Institute, 5 Pawińskiego Str., 02-106 Warsaw, Poland; 4Faculty of Medicine, Lazarski University, 43 Świeradowska Str., 02-662 Warsaw, Poland

**Keywords:** obesity, overweight, bariatric surgery, cost-effectiveness, cost

## Abstract

Weight loss surgery is linked to health benefits and may reduce the cost to the public healthcare systems. The aim of this study was to assess the cost and cost-structure in the one-year periods before and after a bariatric surgery in the Polish nationwide registry. The study included 2390 obese adults which underwent surgical treatment for obesity in 2017. The cost structure and the total costs per patient for one year before bariatric surgery, preoperatively, and for one year after surgery were analyzed. The total cost of the postoperative period was about PLN 3 million lower than during the preoperative period. After bariatric surgery, a reduction of approximately 59% in costs associated with hospital treatment was observed. The costs of outpatient specialist services, hospital treatment, psychiatric care, and addiction treatment also significantly decreased. There was a negative correlation between the changes in the cost of treatment of patients undergoing obesity surgery and their age. The health care cost during the period of one year after bariatric surgery is lower than in the year preceding the surgery (a greater cost difference is observed in younger people). This is mainly influenced by the reduction in costs associated with hospital treatment.

## 1. Introduction

Obesity is defined by the World Health Organization (WHO) as an abnormal or an excessive fat accumulation [1]. The prevalence of being overweight and obesity (assessed in practice as Body Mass Index ≥ 25 and ≥ 30 kg/m^2^, respectively) is increasing in developed and developing countries, as well as in both children and adults. In 2020, being overweight or obese was diagnosed in about 39 million children under the age of five, and in over 340 million children and adolescents (ages 5–19 years) worldwide [2]. The number of obese people has increased three times since 1975, and the prevalence of obesity was 650 million in 2016 [3]. However, epidemiologic assessments have predicted that in 2030, there will be 1.12 billion obese people worldwide [4]. 

It is well known that obesity increases mortality, and it has been shown that in 2010, excess body mass caused 3.4 million deaths worldwide [5]. Obesity decreases the expected lifetime, shortening it by 5–20 years [6]. Being overweight and obese is also related to a considerable increase in comorbidity and high healthcare costs. Such costs increase steadily with the increasing prevalence of obesity. It is estimated that the healthcare costs related to obesity may be up to 40% higher when compared to costs of comparable populations with normal body mass [7]. The systematic review of studies performed in the US showed that the combined cost of being overweight or obese in 2008 in this country was USD 113.9 billion. It is estimated that the direct medical cost of being overweight and obese in the US is approximately 5.0% to 10% of the US healthcare spending [8]. According to the WHO European regional obesity report of 2022, in 2014, obesity was responsible for 8% of the health costs in the EU Member States, and the OECD (The Organization for Economic Co-operation and Development) countries will spend, between 2020 and 2050, an average of 8.4% of their entire annual health budget on treating the consequences of being overweight (including obesity) [9]. 

Considering all the health consequences of obesity, including increased mortality and high health-care expenses, prophylaxis and treatment of obesity, it is becoming one of the most important tasks of research and practice worldwide. Diet, physical activity, and weight management services are very important in dealing with obesity. Lifestyle modification is necessary for successful weight reduction and its maintenance in the longer term. It is well known, however, that compared to non-surgical treatment for obesity (diet, weight-reducing medications, behavioral therapy, or any combination thereof) bariatric surgery is a more effective treatment option for severe obesity not only in terms of weight loss but also with respect to obesity complications [10]. The indications for the surgical treatment of obesity are becoming less stringent. Nevertheless, it should be emphasized that surgery does not exempt patients from lifestyle modifications (which is essential to achieve the expected weight reduction as a result of surgery). According to the American Society for Metabolic and Bariatric Surgery (ASMBS) and the International Federation for the Surgery of Obesity and Metabolic Disorders (IFSO), bariatric surgery is recommended for individuals with a body mass index ≥35 kg/m^2^ (regardless of presence, absence, or severity of co-morbidities) and should be considered for individuals with BMI of 30–34.9 kg/m^2^ and coexisting metabolic disease [11]. Considering the above data, it is not surprising that the number of bariatric surgery procedures is increasing (although the growth rate of surgery per eligible population is lower than the growth rate of the obesity population) [12]. However, it should be noted that surgery is a very expensive procedure. A systematic literature review of published cost analyses showed that the mean total procedural cost (per case) in 2016 was USD 14,389 (obviously, costs vary between countries) [13]. When considering the positive health effects associated with the surgical treatment of obesity compared with the related costs, the question then arises as to whether bariatric surgery entails positive socio-economic implications. It seems that the long-term effects of bariatric procedures on body mass reduction and subsequent cure or reduction of frequency or progression of complications of obesity result in a decrease in the general healthcare expenditures and decreased mortality [14]. However, this effect is undoubtedly delayed, and it is uncertain to what extent the early post-operation costs are decreased. It should also be noted that some studies do not confirm the reduction of overall healthcare costs after bariatric procedures in the long term [15,16]. Klein at al [14] showed that in patients with type 2 diabetes and BMI ≥35 kg/m who underwent bariatric surgery, the cost savings, on average, began to accrue to third-party payers at 3 months and surgery costs were recovered at 30 months. In turn, in the study of Kelles SM et al. [17] regarding the general population of obese patients, the direct costs over the first year after the procedure were greater than before, the patients’ consumption of resources increased and hospitalizations doubled. In the face of these discrepancies, the aim of our study was to assess the cost and cost-structure in the one-year periods before and after a bariatric surgery in the Polish nationwide registry. 

## 2. Materials and Methods

This is a retrospective analysis of the national database. Patients who underwent bariatric surgery between 1 January 2017 and 30 September 2017 were included in the study. The complete list of patients who have undergone bariatric surgery was generated from the National Health Fund (NHF) database. NHF is the state-owned payer of all reimbursed healthcare procedures countrywide; therefore, all services performed as part of the public healthcare must be submitted to and registered in the NHF database. All bariatric surgery procedures are listed under the Diagnosis Related Group F17. For the purpose of this study, only the adult patients’ data were extracted. This group was additionally stratified into 5 subgroups according to age (18–25, 26–40, 41–50, 51–60, ≥61 years of age). Data on sex, postoperative mortality, costs (hospitalizations, medications, specialist care including psychiatric care, primary care, dental treatment, rehabilitation, and others) were analyzed. The analysis was conducted separately for three periods: a pre-operative period (twelve months before the surgery), a perioperative period (from the day of surgery to thirty days after it) and the postoperative period (from the first to the thirteenth month after the surgery). The cost structure in each period was assessed and the mean costs per patient in the pre-operative and post-operative periods were compared.

### Statistical Analysis

Statistical analysis was performed using the R Project for Statistical Computing program (version 3.5.1.). Individual variables were presented using basic descriptive statistics. Nominal variables were described using frequency measures: the number n and % of the group; and quantitative variables were described using measures of central tendency and dispersion. Normality of distribution was tested using the Kolmogorov–Smirnov test, data skewness and kurtosis indices and visual assessment of histograms. Equipotency of subgroups (men’s and women’s group) was checked using the chi-square test which showed that the subgroups were not equipotent. Comparison of preoperative and postoperative costs was performed using the non-parametric Wilcoxon test for paired samples because the assumptions of the parametric test were not met (high skewness and kurtosis of the data, lack of normality of the distribution), assessing the measure of strength of the effect using the binomial correlation coefficient for matched pairs (rc). Comparison of the change in costs after surgery in relation to gender was performed using the non-parametric Mann–Whitney U test because the assumptions of the parametric test were not met (high skewness and kurtosis of the data, lack of normality of the distribution, lack of equality of subgroups), assessing the measure of strength of the effect using the Glass rank biserial correlation coefficient (rg). Given the non-normality of the distribution of the cost data and the presence of outlier observations, analysis of the association between age and cost change was performed using Spearman correlation coefficients (rs). All tests performed were two-sided and a materiality level of 0.05 was used.

## 3. Results

The study comprised 2390 patients (601 men and 1789 women). The mean age of this population was 41.5 ± 10.5 years, whereas the youngest patient was 18 years old, and the oldest was 74 years old. Almost 90% of all patients were between the 26th and 60th years of age, and patients younger or older constituted only about 5%. The detailed age structure is shown in Figure 1. 

From January 2017 to July 2019 (31 months), there were 16 deaths in the studied group of patients, 7 deaths among men and 9 among women. Mortality within the period of 360 days after surgery was 0.25% (6 deaths). In-hospital mortality was very low—only one person died 7 days after surgery. The mean age at death was 48.75 years (SD = 9.32), the youngest patient died at the age of 32, and the oldest at the age of 66. The death occurred on average 15 months (SD = 8.6 months) after surgery (the earliest: 7 days after surgery; the latest: 30 months after surgery).

Considering all analyzed periods, the perioperative period was the most expensive. It generated a cost of PLN 27.30 million (about USD 6 million) and most of it was related to the cost of the bariatric surgery itself (PLN 26.29 million, USD 1.4 6 million), which constituted 96.3% of the total expenditure in this interval. Costs of a stay at the intensive care unit in the perioperative period constituted 0.8%, and the costs of hospital treatment during the period of 30 days after surgery are 1.4% of the total perioperative costs. Costs of pharmacotherapy in the first 30 days after surgery accounted for 1.2% of the total perioperative costs and was generated mainly of low-molecular-weight heparins (0.9%) and proton pump inhibitors (0.2%). Other perioperative costs included blood products (fresh frozen plasma, red cell concentrates) (0.2%), and parenteral nutrition (total or partial) amounting to 0.1% of all perioperative costs in total. Detailed cost structure of the total perioperative costs is presented in Table 1.

With regard to preoperative costs, for the entire group of patients, the total costs of treatment in the year preceding surgery amounted to PLN 6.91 million (USD 1.5 million) and the highest cost was attributed to the hospital treatment (60% of the total cost). The next most costly type of services were medications (15%) and specialist outpatient services (13.2%). The cost of psychiatric care and addiction treatment amounted to 4.1%, and the cost of medical rehabilitation and dental treatment constituted 2.2% of the total costs each. The smallest part of the cost was primary health care services (0.1%) and nursing and care services (0.2%).

The total cost of the postoperative period was PLN 4,104,489.88 (USD 912,108.866) which was about PLN 3 million (USD 667,000) lower in comparison with the preoperative period. Similar to preoperative costs, the majority of expenses were related to hospital treatment (40% of postoperative costs), medications (24%) and outpatient specialist services (15%). Medical rehabilitation and psychiatric care together with treatment of addictions generated 11.2% of total cost in postoperative period. 

The comparison of the structure of preoperative and postoperative costs showed the biggest differences in the percentage of costs related to hospital treatment and medications. The percentage of costs related to hospital treatment decreased in the postoperative period (60% of the total preoperative costs vs. 43% of the total postoperative costs), while the percentage of costs related to medications increased (15% of the total preoperative costs vs. 27% of total postoperative costs). The detailed structure of pre- and postoperative costs in the study group is presented in Table 2.

The analysis of the mean costs per patient showed that the total cost one year after surgery was significantly lower than that before surgery (median difference −716.71 PLN [CI −849.11; −594.03], *p* < 0.001 this equals roughly −159 USD). The costs of outpatient specialist services, hospital treatment, psychiatric care, and addiction treatment significantly decreased. A comparison of the preoperative and postoperative mean costs is shown in Table 3.

No statistically significant difference was found between men and women, for the total treatment cost and for individual types of services, in mean change in costs after surgery compared to the period before surgery. 

The analysis of the relationship between the changes in the cost of treatment of patients undergoing obesity surgery and their age showed a negative correlation. The older the patients were, the smaller was the statistically significant change in the total cost, and cost of outpatient specialist services, hospital treatment, preventive health programs and medications (Table 4).

## 4. Discussion

Based on the available data, this study on a population of patients diagnosed with obesity was the first to evaluate and compare direct health care costs before and after bariatric surgery using the nationwide data from a Polish registry. The study indicated an over 40% reduction in health-care costs in the one-year period after bariatric surgery. This is an important observation from the point of view of the medical service payers when it comes to the long-term planning of tasks and financing of individual services. Our findings are consistent with a study conducted in Brazil by Sussenbach et al. [18]. This cohort study showed a reduction of approximately 32% in the expenditure incurred on medications, additional examinations and medical appointments during the first 12 months after bariatric surgery. Moreover, during the subsequent period of time, a reduction of costs was further observed, and it was progressively greater (within 24 months, a reduction of about 58% and 74% at 24 and 36 months, respectively). Unfortunately, our study does not include monitoring longer than 12 months after bariatric surgery. Another Brazilian study published by Turri JAO et al. this year [19], estimated patients’ direct health-care costs referring to bariatric surgery in a single center during a period of 6 months pre-intervention and 6 months post-intervention. It was shown that mean direct costs of hospitalization, imaging, and medication decreased after surgery. However, total direct costs, consultations, and laboratory exams increased after the intervention. Additionally, the study by Bøgelund M et al. [20] performed in Denmark revealed that in people with obesity receiving bariatric surgery, healthcare costs significantly and persistently increase during the first five years after surgery (especially in the first year after intervention). Another analysis using real-world data from national registries concluded that there are no cost savings of bariatric surgery in the short run (3 years after the procedure) [21]. In turn, a meta-analysis of 61 studies examining the 23 years period has shown that bariatric surgery is largely cost-saving, mainly due to the reduction of costs of medication [22]. In our study, the total costs of medications decreased one year after surgery; however, there is some discrepancy between the medication costs per patient expressed as a median and mean (we will perform an additional analysis of this issue). Mean values may be artificially high due to outliers (high costs of an individual patient). For this reason, the medians seem to be a better parameter. It should be noted that we use the median of differences (more precisely, the so-called ‘pseudomedian’—the value calculated by the Wilcoxon test), which does not coincide with the usual arithmetic difference (mean after the operation minus the mean before the operation). Additionally, the period of our observation is short, and some obesity complications requiring the use of medication may not have been resolved after a year. 

The discrepancies in the literature regarding the impact of bariatric surgery on health care costs may be due to several reasons. Firstly, the studies differ significantly in methodology as well as in the size of the assessed population. Secondly, the incidence of post-procedural complications may vary significantly from country to country. Additionally, post-bariatric care costs may vary between countries depending on the follow-up schemes used. It is known that patients after bariatric surgery require follow-up medical appointments, monitoring of nutritional deficiencies, and use of specific medications; however, the frequency of visits and the ordered laboratory tests may significantly differ depending on the dominant types of operations performed in a given country. Unfortunately, in Poland, there are very few centers offering integrated care for obese patients, and many patients after bariatric surgery are supervised by primary care physicians only.

In Poland, bariatric surgery procedures have been reimbursed by NHF since 2017 and the reimbursed sum ranges from 10,944 to 11,723 PLN per patient (this is the amount the NHF pays the hospital for a patient’s hospital stay, surgery, medications, etc.), which equals roughly EUR 2300 or USD 2400. The cost of the procedure is much lower than in other European countries or in the US [13], although this is a reflection of the lower income and lower general prices in Poland. Regardless, the reimbursement has facilitated general access to those procedures.

The analysis in this study showed a significant predominance of women (75%) undergoing bariatric surgery. Similarly, other published studies are characterized by a significant predominance of women [23,24]. The above scheme may indicate that women are more aware and knowledgeable about obesity and the need to treat it. In addition, there is a widespread trend among women to take care of themselves and their holistic well-being. The predominance of women among patients undergoing bariatric surgery does not affect the cost of care, however. In our study, no statistically significant difference was found between men and women in the mean change in treatment costs after surgery. 

The problem of obesity also affects the elderly population. In people aged 60 years and older, 37.1% of men and 33.6% of women are classified as having obesity (based on a body mass index (BMI) ≥ 30 kg/m^2^) [25]. Although weight loss interventions in older adults remain a controversial topic, some people may benefit from them [26]. Bariatric surgery offers obese elderly patients a satisfactory result, and it can improve their quality of life [27] with an acceptable surgical risk [28]. According to this year’s newly updated indications for metabolic and bariatric surgery, there is no evidence to support an age limit for patients seeking bariatric surgery. The most important is a careful selection that includes an assessment of frailty [11]. There are also studies pointing to bariatric surgery as cost-saving even in the older population. In the study by Marihart CL et al. [29], estimated savings started accruing within three months of surgery. However, it is known that the survival time of older people is significantly shorter than that of young people. In addition, younger patients have a lower burden of comorbidities both pre- and post-operatively. So, in the long run, cost saving seems probably greater in younger people. In our study, people older than 60 years old constituted 5% of the research group. We showed a negative correlation between the changes in the cost of treatment of patients undergoing obesity surgery and their age (the older the patients were, the smaller the change in the total cost). 

In summary, this is the first analysis of a national database to assess the cost and cost structure in the one-year periods before and after bariatric surgery in the Polish population. The strength of this study lies in the high number of analyzed surgeries, making the results more reliable than those obtained in smaller studies. On the other hand, the study has some limitations. The first one is its retrospective character. As a result of this, we had no information on the type of operation, obesity categories, and comorbidities. Additionally, we did not have data on other methods of treating obesity so the comparison of costs and effectiveness of bariatric surgery with non-surgical treatment was impossible. The choice of services which were analyzed in the cost structure was limited to the information available in the database. Additionally, we could only analyze data on operations reimbursed by the state-owned payer while some procedures may also be performed in private centers. Data on dental treatment are also probably incomplete, as most dental procedures in Poland are not paid for by the National Health Fund. Another limitation is the short observation period (one year before and after the operation), mainly related to the COVID pandemic, which, as is known, significantly affected elective surgeries and medical check-ups. The last limitation is the lack of a control group. The relatively short duration of the study and the lack of a control group limited the study’s capacity to conclude that bariatric surgery is a cost-saving method of obesity treatment. Therefore, this study is unable to fully inform decision-makers of the value of bariatric surgery. It should be noted, however, that the National Health Fund database of Poland is the biggest healthcare database in Poland, which registers all services performed as part of public healthcare. Therefore, it can be assumed that the costs reported in this study directly represent the country’s spending on bariatric surgery, and obviously, the reliability of conclusions based on an observation of a large group of patients is high. Additionally, the NHF database allows us to look into the distribution of the costs, showing the separate costs of the different procedures (such as hospitalizations, ambulatory care, and rehabilitation) and cost of the medications.

## 5. Conclusions

This analysis shows that in people with obesity who underwent bariatric surgery, the use of health care and its costs up to one year after bariatric surgery is lower than in the year preceding the surgery (the greater cost differences are observed in younger people compared to older ones). This is mainly influenced by the reduction in costs associated with hospital treatment. Further research with a longer period of observation, cost-effectiveness modeling, or with a control group is needed to clarify if bariatric surgery is a cost-saving method when treating obesity.

## Figures and Tables

**Figure 1 ijerph-20-01118-f001:**
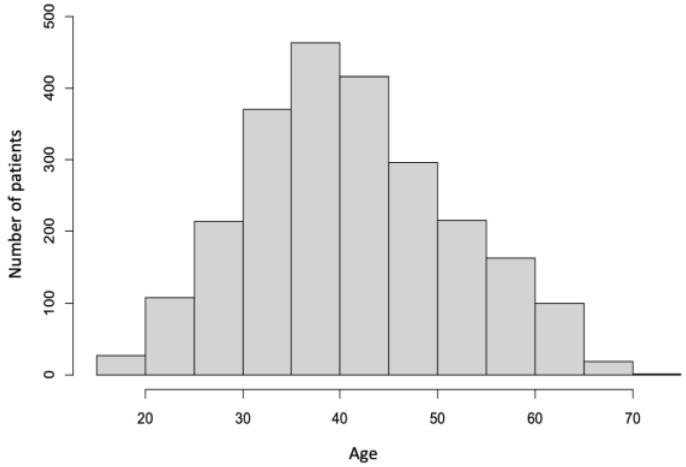
Age structure of the study group.

**Table 1 ijerph-20-01118-t001:** Structure of total perioperative costs in the study group.

Type of Service	Value, PLN	% of All Costs
Surgery (hospital stay, medication, surgical procedure, etc.)	26,294,985	96.3
Hospital treatment within 30 days of surgery	383,418	1.4
Low-molecular-weight heparins	240,645	0.9
Stay in the intensive care unit	228,381	0.8
Transfusions	56,146	0.2
Proton pump inhibitors	42,668	0.2
Parenteral nutrition	32,452	0.1
Other medications	19,632	0.1
Separately contracted services	2786	<0.1
Other	1888	<0.1
Outpatient specialist services	1241	<0.1
Psychiatric care and addiction treatment	299	<0.1
Total perioperative costs	27,304,540	100

To convert the values in PLN to USD they should be divided by 4.4 (USD 1 = PLN 4.4).

**Table 2 ijerph-20-01118-t002:** Structure of total pre- and postoperative costs in the study group.

	Preoperative Costs		Postoperative Costs	
Type of Service	Value, PLN	% of All Costs	Value, PLN	% of All Costs
Hospital treatment	4,147,993.50	60.0	1,648,744.54	40.2
Medications	1,033,403.81	15.0	987,204.02	24.1
Outpatient specialist services	913,621.46	13.2	631,268.57	15.4
Psychiatric care and addiction treatment	282,597.00	4.1	231,258.82	5.6
Separately contracted services	212,248.69	3.1	230,548.57	5.6
Therapeutic rehabilitation	152,416.79	2.2	168,277.08	4.1
Dental treatment	151,247.11	2.2	143,501.02	3.5
Nursing and care services	12,318.32	0.2	46,878.14	1.1
Primary health care	3856.00	0.1	13,166.13	0.3
Preventive health programs	28.00	<0.1	3523.00	0.1
Pilot programs	-	-	120.00	<0.1
Total costs	6,909,730.67	100.0	4,104,489.88	100.0

To convert the values in PLN to USD they should be divided by 4.4 (USD 1 = PLN 4.4).

**Table 3 ijerph-20-01118-t003:** A comparison of the preoperative and postoperative mean costs (in PLN) per patient in the study group.

	Preoperative Costs		Postoperative Costs			
Type of Service	Median (Min-Max)	Mean ± SD	Median (Min-Max)	Mean ± SD	MD (95% CI)	*p*
Outpatient specialist services	252.00 (0.00–3702.76)	382.27 ± 430.94	146.25 (0.00–3029.67)	264.13 ± 341.64	−99.28 (−112.66; −86.33)	<0.001
Dental treatment	0.00 (0.00–1970.59)	63.28 ± 175.39	0.00 (0.00–2911.00)	60.04 ± 183.26	−14.61 (−29.27; −0.11)	0.047
Hospital treatment	0.00 (0.00–99,825.60)	1735.56 ± 4281.59	0.00 (0.00–113,665.76)	689.85 ± 5163.06	−2418.00 (−2600.00; −2224.04)	<0.001
Psychiatric care and addiction treatment	0.00 (0.00–38,863.74)	118.24 ± 997.17	0.00 (0.00–13,654.67)	96.46 ± 687.63	−78.99 (−97.50; −61.87)	<0.001
Primary health care	0.00 (0.00–527.00)	1.61 ± 17.71	0.00 (0.00–638.00)	1.47 ± 18.54	−0.14 (−1.14; 0.86)	0.472
Preventive health programs	0.00 (0.00–28.00)	0.01 ± 0.57	0.00 (0.00–2025.00)	5.51 ± 69.08	634.38 (509.87; 738.38)	<0.001
Therapeutic rehabilitation	0.00 (0.00–8113.20)	63.77 ± 320.92	0.00 (0.00–10,160.00)	96.76 ± 598.92	9.50 (−29.12; 48.62)	0.624
Separately contracted services	0.00 (0.00–67,127.64)	88.81 ± 2327.01	0.00 (0.00–63,280.18)	70.41 ± 1926.56	−703.50 (1725.54; 271.32)	0.224
Medications	204.32 (0.00–11,194.52)	432.39 ± 710.96	315.34 (0.00–20,705.37)	413.06 ± 920.21	93.34 (82.61; 104.00)	<0.001
Total costs	1254.11 (0.00–100,448.10)	2891.10 ± 5374.82	693.55 (0.00–114,325.51)	1717.36 ± 6298.87	−716.71 (−849.11; −594.03)	<0.001

Abbreviations: MD (95% CI) median difference in postoperative cost minus operative cost with 95% confidence interval; SD standard deviation; To convert the values in PLN to USD they should be divided by 4.4 (USD 1 = PLN 4.4).

**Table 4 ijerph-20-01118-t004:** Correlation between age and the change in the cost of postoperative treatment compared to preoperative treatment.

Type of Service	Spearman’s Correlation Coefficient (*r_s_*)	*p*
Outpatient specialist services	−0.10	<0.001
Dental treatment	−0.02	0.365
Hospital treatment	−0.07	0.001
Psychiatric care and addiction treatment	−0.04	0.081
Primary health care	−0.01	0.668
Preventive health programs	−0.08	<0.001
Therapeutic rehabilitation	0.02	0.298
Separately contracted services	−0.02	0.349
Medications	−0.08	<0.001
Total costs	−0.10	<0.001

## Data Availability

The data presented in this study are available on reasonable request from the corresponding author.

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
