# Peer review of "Cost of Surgical Treatment of Obesity and Its Impact on Healthcare Expense–Nationwide Data from a Polish Registry"

_ijerph, 2023, doi:10.3390/ijerph20021118_

Round 1

Reviewer 1 Report

1. Please, explain the difference between hospital costs and in-hospital treatment costs  mentioned twice in the abstract

2. Please complete reference  9,  adding page and/or table number  in  GUS report including  a surprising  estimation of  15  billion PLN obesity indirect cost  in Poland

3. Please  justify the use of cost minimisation analysis method / explain why no other metod of economic analysis could be used

Author Response

We would like to thank the Reviewer for the opportunity to revise the manuscript, as well as for the helpful comments and suggestions. Please find below a point-by-point response to the comments. Any revisions to the manuscript are marked up using the “Track Changes” function.

  1. Please, explain the difference between hospital costs and in-hospital treatment costs mentioned twice in the abstract

There is no difference between "hospital costs" and "in-hospital treatment costs" mentioned in the abstract. The nomenclature has been unified.

  1. Please complete reference 9, adding page and/or table number in GUS report including a surprising estimation of 15 billion PLN obesity indirect cost in Poland

Thank you for this comment. We have performed an extensive overview of published data on obesity costs in Poland, and we have come to the conclusion that none of them are based on reliable sources. In this situation, we rewrote this fragment of the introduction section giving WHO data for Europe.

  1. Please justify the use of cost minimisation analysis method / explain why no other method of economic analysis could be used

We tried to look at the problem from the point of view of the payer which uses the cost-minimization approach. Additionally, the NHF database contains only limited data, the observation period is rather short, and based on the database we were not able to create a well-matched control group. Therefore, we have chosen this method.

Reviewer 2 Report

- Please, to mention the data in dollars in addition to the value of the local currency

- Line 54 to 56:” It is well known that bariatric surgery seems to be the most effective treatment of severe obesity in terms of weight loss, but it is also more effective with respect to obesity complications [10]”. It is suggested to mention this probable effectiveness compared to what other treatments? For example, nutritional and dietary treatment?

- A bias that the study presents and that the authors do not clarify, or mention is nutritional and dietary treatment, its costs and effectiveness compared to surgery, in addition to the fact that the former entails nutritional education and changing lifestyles. In the entire proposal, only surgery is mentioned as the only alternative.

- The material and methods section does not specify the type of study, the type of test, the logistics, and the control of variables.

- In the statistics, although the key variables to be measured are addressed, but they are not specific, such as the effectiveness of surgery and its costs and compared with the treatments that are offered or are in the health regulation in Poland, such as nutritional treatment, as well from any other that is suggested, always based on evidence of failure of treatment with diet, exercise, and nutritional counselling.

- Registration before an Ethics Committee is not mentioned.

- The criteria for selection, non-selection, control variables, etc. are not mentioned.

- The statistical analysis does not integrate by age group or condition of overweight or obesity in addition to other comorbidities.

- Although the authors mention the limitations, they do not mention the selection biases, systematic in addition to considering the cost compared to another treatment that is known to have better efficacy such as diet, exercise and that it would be important for the authors to mention, what management is usually given to these patients in addition to surgery and what criteria are followed to opt for surgery. Therefore, the results lack the social value of what is sought, since the methodological quality was not controlled to be able to see the costs with this and other forms of management.

- Diet, exercise, present comorbidities are not mentioned, or considered in addition to what has already been mentioned.

- Given the scientific evidence with the benefits of diet, exercise and counseling, it is not only option a surgery without a clinical, regulatory and scientific justification for Poland, as is the case.

Author Response

We would like to thank the Reviewer for the opportunity to revise the manuscript, as well as for the helpful comments and suggestions.

Initially, it should be emphasized that our study is a retrospective analysis whose design is driven by the content of the NHF database. It contains only limited data. We did not have data on height, weight, BMI, or how obesity was treated (diet, physical activity, and medication that were used were not included in the database we used). Therefore, we were not able to construct a matched control group with patients who underwent other obesity treatments, or even a control group that would include obese patients. In this situation, only patients who have undergone bariatric surgery were included in the study. We performed the analysis of the structure of costs of NHF and total costs per patient with obesity for one year before bariatric surgery, preoperatively, and for one year after surgery (this is no regression-type study).

Please find below a point-by-point response to the comments. Any revisions to the manuscript are marked up using the “Track Changes” function.

- Please, to mention the data in dollars in addition to the value of the local currency

Data in dollars was added to the text. We did not change the tables. The additional data makes them hardly readable. Below the tables, however, we have provided the conversion rate of PLN to USD.

- Line 54 to 56:” It is well known that bariatric surgery seems to be the most effective treatment of severe obesity in terms of weight loss, but it is also more effective with respect to obesity complications [10]”. It is suggested to mention this probable effectiveness compared to what other treatments? For example, nutritional and dietary treatment?

The non-surgical comparison was added to the text (line 65)

- A bias that the study presents and that the authors do not clarify, or mention is nutritional and dietary treatment, its costs and effectiveness compared to surgery, in addition to the fact that the former entails nutritional education and changing lifestyles. In the entire proposal, only surgery is mentioned as the only alternative.

As mentioned at the beginning, this is a retrospective analysis of the database, which focused on surgery. We did not have data on other methods of treating obesity so we couldn't make comparisons with non-surgical treatment. We have added this information to the study limitations (line 301-304)

- The material and methods section does not specify the type of study, the type of test, the logistics, and the control of variables.

We have added the information on the study type to the manuscript (line 99, section 2.1). The information about all statistical tests used in the study was specified in the 2.1 section. This is no regression-type study, and there is no control group in the study.

- In the statistics, although the key variables to be measured are addressed, but they are not specific, such as the effectiveness of surgery and its costs and compared with the treatments that are offered or are in the health regulation in Poland, such as nutritional treatment, as well from any other that is suggested, always based on evidence of failure of treatment with diet, exercise, and nutritional counselling.

As mentioned above, the study focused only on surgery. In the analysis, we used variables that were available in the database. Because of the lack of data, the comparison with non-surgical treatment was impossible.

- Registration before an Ethics Committee is not mentioned.

As mentioned in the Institutional Review Board Statement (line 333-335), the ethical review and approval were waived for this study due to the retrospective and non-invasive nature of the study.

- The criteria for selection, non-selection, control variables, etc. are not mentioned.

The only selection criterion was a performed bariatric surgery, all patients who have undergone this type of operation have been included (line 99-100). We used the variables that were available in the database. This information is mentioned in the limitations of the study. The study assesses only the costs during particular time periods. This is a no regression-type study. We did not assess relationships between variables so there was no need to use controlling variables.

- The statistical analysis does not integrate by age group or condition of overweight or obesity in addition to other comorbidities.

The results of the analysis of the relationship between the changes in the cost of treatment of patients undergoing obesity surgery and their age are described in the last paragraph of the results section (line 204-211, table 4.). Because lack of data, we couldn't perform an analysis by the obesity categories, and comorbidities (301-302). We have added this information to the study limitations.

- Although the authors mention the limitations, they do not mention the selection biases, systematic in addition to considering the cost compared to another treatment that is known to have better efficacy such as diet, exercise and that it would be important for the authors to mention, what management is usually given to these patients in addition to surgery and what criteria are followed to opt for surgery. Therefore, the results lack the social value of what is sought, since the methodological quality was not controlled to be able to see the costs with this and other forms of management.

We added to the paragraph on study limitations the information about the lack of data regarding non-surgical methods of treating obesity (what makes the comparison of costs and effectiveness of bariatric surgery with non-surgical treatment impossible) – line 302-304. We didn't have the information on what management was given to these patients in addition to surgery and what criteria were followed to opt for surgery. The aim of our study wasn’t the comparison of the costs of bariatric surgery with non-surgical obesity treatment. We evaluated only the medical care costs before and after surgery.

- Diet, exercise, present comorbidities are not mentioned, or considered in addition to what has already been mentioned.

As mentioned at the beginning, we do not have data on diet, exercise, present comorbidities.

- Given the scientific evidence with the benefits of diet, exercise and counseling, it is not only option a surgery without a clinical, regulatory and scientific justification for Poland, as is the case.

We don't suggest that surgery is the only treatment option for obesity. We do not question the huge role of diet, exercise, and counseling in obesity treatment. We have added this information to the introduction (line 61-70). 

Reviewer 3 Report

This study assessed the costs and cost structure in the one-year period before and after a bariatric surgery using data from the Polish nationwide registry. Whether bariatric surgery among patients with obesity can reduce medical costs for this patient population is of high interest among researchers and payers, therefore this study tried to address an important research question and may have implications on reimbursement decisions by payers and health systems. However, the study design is relatively effortless which limited its capability to conclude the cost reduction due to bariatric surgery, as the study only looked at a relatively short time period (1 year) and is lack of a control group. Therefore, I don't think this study is able to fully inform decision makers on the value of bariatric surgery. 

In order to assess the implications on costs and health outcomes of a certain intervention, cost-effectiveness modeling is now the standard method adopted by the majority of researchers. Advantages of cost-effectiveness modeling are that it is able to follow patients over a long period, even through the whole lifetime, to collect the long-term costs and outcomes. Furthermore, a control group is usually included in the model to help determine the effects caused by the intervention. It is also able to synthesize the effects on both costs and patient outcomes of one intervention in one measurement. I wonder if the authors thought about taking this modeling approach when they designed the study, as this method is superior to the current study design. 

Alternatively, if the focus of the study is solely on costs instead of health outcomes, it is acceptable to utilize a retrospective cohort study design similar to the current design, however, the authors need to have a control group in which patients with the same indication did not receive the surgery. Without a control group, it is not appropriate to conclude the pre-post difference observed is caused by the surgery, as the patients who get surgery may be very different from those without surgery. It is essential to make sure the treatment and control group are comparable, some common approach include propensity score matching to match treated patients with similar control patients. 

Other comments:

In introduction, the authors did not review the previous related literature and fully discuss where is the gap and how the current study adds to literature. the authors did mention several studies in Discussion, but they did not discuss the reasons for mixed results from previous literature. 

English language is awkward throughout the manuscript and needs substantial editing. 

Author Response

We would like to thank the Reviewer for the opportunity to revise the manuscript, as well as for the helpful comments and suggestions. Please find below a point-by-point response to the comments. Any revisions to the manuscript are marked up using the “Track Changes” function.

This study assessed the costs and cost structure in the one-year period before and after a bariatric surgery using data from the Polish nationwide registry. Whether bariatric surgery among patients with obesity can reduce medical costs for this patient population is of high interest among researchers and payers, therefore this study tried to address an important research question and may have implications on reimbursement decisions by payers and health systems. However, the study design is relatively effortless which limited its capability to conclude the cost reduction due to bariatric surgery, as the study only looked at a relatively short time period (1 year) and is lack of a control group. Therefore, I don't think this study is able to fully inform decision makers on the value of bariatric surgery. 

We agree with the reviewer and we have added this information to the study's limitations.

In order to assess the implications on costs and health outcomes of a certain intervention, cost-effectiveness modeling is now the standard method adopted by the majority of researchers. Advantages of cost-effectiveness modeling are that it is able to follow patients over a long period, even through the whole lifetime, to collect the long-term costs and outcomes. Furthermore, a control group is usually included in the model to help determine the effects caused by the intervention. It is also able to synthesize the effects on both costs and patient outcomes of one intervention in one measurement. I wonder if the authors thought about taking this modeling approach when they designed the study, as this method is superior to the current study design. 

Alternatively, if the focus of the study is solely on costs instead of health outcomes, it is acceptable to utilize a retrospective cohort study design similar to the current design, however, the authors need to have a control group in which patients with the same indication did not receive the surgery. Without a control group, it is not appropriate to conclude the pre-post difference observed is caused by the surgery, as the patients who get surgery may be very different from those without surgery. It is essential to make sure the treatment and control group are comparable, some common approach include propensity score matching to match treated patients with similar control patients. 

We fully agree with the reviewer, but it should be noted the NHF database contains only limited data (the database is created on the basis of data that have to be reported by medical facilities to the NHF). We have no information on patients' comorbidities and it's hard to evaluate particular health outcomes after bariatric surgery in the study group. The data focus only on costs. The observation period is also short (as mentioned in the manuscript because of the pandemic). We are greatly interested in carrying out cost-effectiveness modeling but we believe we don't have enough data at this time.

Unfortunately, for the same reason, we are unable to create a comparable, propensity score-matched control group (the database contains no data like BMI, concomitant conditions, etc.).

Considering this valuable comment of the reviewer we have reworded the conclusions.

Other comments:

In introduction, the authors did not review the previous related literature and fully discuss where is the gap and how the current study adds to literature. the authors did mention several studies in Discussion, but they did not discuss the reasons for mixed results from previous literature. 

In fact, data from the previously published literature we have mainly presented in the discussion section (to not duplicate contents). According to reviewer suggestions, we have extended the discussion (discussing gaps and discrepancies) in the introduction section. The reasons for mixed results from previous literature were shortly discussed at the end of the first paragraph however, it may not have been legible. So, we expanded it a bit and moved it to a separate paragraph.

English language is awkward throughout the manuscript and needs substantial editing. 

The manuscript underwent English revisions by native speakers.

Round 2

Reviewer 3 Report

Thank you for considering my comments. I understand it is impossible to carry out analysis with a comparison group or cost-effectiveness analysis due to lack of data. Although the methods are not superior, this study still is potential valuable given the authors are using the National Health Fund database of Poland which registered all services performed as part of the public healthcare. Therefore, the costs reported in this study directly represented the country's spending on this service. I think in the Discussion section, the authors should highlight more on the advantage of utilizing NHF data. 

Author Response

Thank you for these additional comments. According to the reviewer's suggestion, in the Discussion section, we have highlighted more of the advantages of utilizing NHF data (lines 315-322).